# Costly mistakes: Why and when spelling errors in resumes jeopardise interview chances

**Philippe Sterkens[1]\*, Ralf Caers[2], Marijke De Couck[3,4], Victor Van Driessche[1], Michael Geamanu[1], Stijn Baert[1,5,6,7,8]**

**1** Ghent University, Ghent, Belgium, **2** KU Leuven, Leuven, Belgium, **3** Vrije Universiteit Brussel, Brussel, Belgium, **4** Odisee University College, Bruxelles, Belgium, **5** University of Antwerp, Antwerpen, Belgium, **6** Université Catholique de Louvain, Ottignies-Louvain-la-Neuve, Belgium, **7** Institute for Labor Economics (IZA), Bonn, Germany, **8** Global Labor Office (GLO), Brussels, Belgium

\* Philippe.Sterkens@UGent.be

## Abstract

The analysis of hiring penalties due to spelling errors has been restricted to white-collar occupations and error-laden resumes. Moreover, the mechanisms underlying these penalties remained unclear. To fill these gaps, we conducted a scenario experiment with 445 recruiters. Compared to error-free resumes, hiring penalties are inflicted for error-laden resumes (18.5 percent points lower interview probability) and resumes with fewer errors (7.3 percent points lower interview probability). Furthermore, we find heterogeneity in penalties inflicted. Half of the penalty can be explained by the perceptions that applicants making spelling errors have lower interpersonal skills (9.0%), conscientiousness (12.1%) and mental abilities (32.2%).

## 1. Introduction

Resume review is a first and crucial step in the hiring process. Its major purpose is to screen out applicants who do not fit the job requirements [1]. During this early screening phase, employers infer otherwise unobservable applicant characteristics (e.g., work ethic) from resumes [2]. Hence, it is in applicants' best interest to positively shape employers' inferences in the short time (about 45 seconds [3]) employers skim through a resume [4]. Applying the seminal signalling theory to resume screening [5, 6], employers ('signal receivers') lack a priori knowledge of applicants' ('signal senders') productivity; hence, there is information asymmetry. Applicants attempt to reduce this asymmetry by compiling resumes with information on characteristics such as education to signal their qualities as potential employees and convincing employers of their suitability [7]. The rationale behind education as a signal is that applicants of lower quality would be unable to complete education programs, thus indicating that graduates possess job skills that applicants without a degree lack [7, 8].

Education sections on resumes are deliberate signals transmitted by applicants. However, unintentional negative signalling, that is, often unintended consequences of sender actions

**Data Availability Statement:** The data and analysis files are available via: https://osf.io/6mevd/.

**Funding:** we did not receive specific funding for this work.

**Competing interests:** The authors have declared that no competing interests exist.

that signal undesirable characteristics, has received relatively little attention in the literature [5]. In the current study, we focus on one instance of unintentional negative signalling in the hiring process, namely poor linguistic care through spelling errors in resumes. We argue that in a hiring context, poor language care can be considered both a strong and visible signal [9] of job-relevant characteristics (infra), which affects hiring decisions. First, spelling errors are potentially strong signals as they contain important information on unobservable characteristics relevant to hiring decisions. More specifically, in prior research (primarily through surveys), employers indicated that they attach value to the written communication of their prospective employees [10–12]. These skills are even more critical because employees are often found to possess inadequate written communication skills [10, 11, 13]. Spelling accuracy could be an increasingly relevant signal to differentiate applicants, thus creating a competitive advantage for applicants with error-free resumes. Second, spelling errors can be a visible signal. That is, signal receivers can easily notice the signalling because of its objective nature–spelling is either correct or not–and readers can easily recognise errors [14].

Having described the potential of spelling errors in resumes to act as unintentional (negative) signals, we now discuss, based on a review of the literature, potential inferences drawn from spelling errors in resumes. These inferences include unobservable applicant characteristics signalled by applicants based on the recruiters' attributions of these errors. Hence, we found three broad domains of potential inferences drawn from spelling errors. First, Martin-Lacroux [15] theorised that applicants with erroneous spelling might be perceived as lacking professionalism, politeness, language and cultural skills and maturity in communication. These *interpersonal skills* are known to have an impact on recruiters' hiring decisions [7, 16] because they reflect an individual's ability to manage complex interpersonal relationships in the workplace [17].

Second, employers could attribute spelling errors to the applicant's personality. These applicants are, in particular, accused of laziness, untidy work and little proofreading [15, 18], all behaviours associated with lower levels of *conscientiousness*. In contrast, conscientious individuals are likely to be hired because they are perceived as goal-oriented, hard-working and loyal workers [19, 20]. Indeed, as a personality trait, conscientiousness is positively associated with job performance [20] and earnings [21]. Third, employers may also attribute spelling errors to applicants' *mental abilities*. This might especially be the case when applications contain multiple errors [15, 22]. Workers with higher mental abilities are attractive applicants for recruiters because they might perform tasks quickly and efficiently. Indeed, measures of general mental abilities are the single best predictor of job performance [23, 24].

In line with the discussed theoretical reasons for a penalty of spelling errors in hiring, earlier empirical studies found that error-laden resumes do indeed make poor impressions on recruiters screening for jobs with high educational requirements. An overview of the empirical literature on effects of spelling errors in resumes is presented in Table 1 below. More specifically, spelling errors in resumes are associated with lower applicant ratings, hiring chances and proposed starting salaries [14, 15, 25–27]. Moreover, there is some initial evidence for heterogeneity of spelling errors' effect with a higher penalty inflicted on more experienced [27] and non-white applicants [28]. In their study, however, Martin-Lacroux and Lacroux [27] also suggest that penalties for spelling errors vary with the recruiter's own spelling capacities, with the penalty disappearing among recruiters with low spelling abilities.

Regarding empirical design, we discover two gaps in these studies that limit their external validity, which is also acknowledged by Martin-Lacroux and Lacroux [27]. First, as illustrated in Table 1, the number of spelling errors featured in pasts experiments' resumes is always substantial, with no less than five errors per application. However, to truly capture a broad range of realistic occurrences of spelling errors in resumes ('ecological validity'), we are convinced

**Table 1. Spelling errors during resume screening: Literature review.**

| Study | Country | Method | Occupation(s) | Main results (on spelling errors) |
|---|---|---|---|---|
| Charney & Rayman (1989) | United States | Survey experiment with 18 recruiters, each evaluating 72 fictitious graduate resumes | Mechanical engineers | • Resumes with errors (average eight to ten) received lower ratings on desire to interview. |
| Charney, Rayman & Ferreira-Buckley (1992) | United States | Study 1: Survey experiment with 47 recruiters, each evaluating 36 fictitious graduate resumes | Marketeer | • Study 1: Resumes with errors (average five to nine) received lower ratings on desire to interview. |
| | | Study 2: Survey experiment with 42 undergraduate students, each evaluating 36 fictitious graduate resumes | | • Study 2: Resumes with errors (average five to nine) received equal ratings on deserving a job interview. |
| van Toorenburg, Oostrom & Pollet (2015) | Netherlands | Survey experiment with 73 recruiters, each evaluating six resumes | HR specialist | • Resumes with errors (five) received lower hireability ratings. |
| Martin-Lacroux (2017) | France | Verbal protocol analysis of 20 recruiters, each evaluating six resumes and cover letters | Banking account manager | • Resumes with spelling errors (ten) are described differently from resumes without errors and resumes with typographical errors (ten). Resumes with spelling errors had the highest rejection rate. |
| | | | | •Spelling errors are attributed to soft skills, abilities and culture. |
| Martin-Lacroux & Lacroux (2017) | France | Survey experiment with 536 recruiters, each evaluating four resumes and cover letters | Banking account manager | • Resumes with spelling errors (five or ten) had the highest rejection rates. |
| | | | | • Only recruiters with considerable spelling abilities penalised error-laden resumes. |
| Shore, Tashchian & Forrester (2021) | United States | Survey experiment with 164 respondents experienced at hiring, each evaluating one fictitious LinkedIn profile ('resume') | Sales manager | • Resumes with spelling errors (eight) received lower interview probabilities, hiring probabilities, salary offers and were perceived more negatively. |
| | | | | • Resumes with spelling errors (eight) of non-white applicants received lower salary offers than resumes of white applicants with spelling errors. |

Notes. Verbal protocol analysis is a qualitative research method that involves asking participants to think aloud while conducting a task. Verbalisations are then transcribed and analysed via a coding scheme. For a definition of 'survey experiments', we refer to Section 2.

that an investigation of cases with lower numbers of spelling errors is an imperative addition to the literature. Second, and perhaps even more important, is that prior studies on the effects of linguistic care in solicitations have exclusively focussed on so-called 'white-collar' jobs. When spelling errors in resumes signal lower interpersonal skills and conscientiousness in addition to mental ability, they could also jeopardise hiring chances in 'blue-collar' jobs where mental abilities are typically less central job demands than in white-collar jobs.

In the present study, we address these concerns by means of a state-of-the-art factorial survey experiment that is a substantial extension of previous designs in multiple aspects. In the experiment, we collect hireability ratings and perceptions about applicants based on resumes with systematically assigned spelling errors, including a lower number of errors (two) than used in earlier literature. A total of 445 real-life recruiters–a significant number in comparison to the literature–partake in the experiment, with a total of 1,335 resumes appraised. Importantly, these recruiters are randomly distributed across eight fictitious vacancies, including both white- and blue-collar jobs. Besides increasing the external validity of our experiment, this randomisation also allows us to investigate heterogeneity in the spelling error effect by occupation characteristics. In addition, our experimental design allows us to investigate the moderating potential of other applicant (specific resume content) and recruiter characteristics (e.g., language sensitivity) not yet addressed in the literature. Finally, we also have recruiters rate applicants by a broad spectrum of candidate perceptions related to interpersonal skills, conscientiousness and mental abilities. Doing so enables us to uncover the mechanisms

underlying the spelling error penalty more comprehensively than previous studies. Whereas the rigour of vignette experiment is at the core of this study's strengths, the hypothetical nature of participants' decisions is our methodology's main limitation.

## 2. Method

Factorial survey experiments enable the analysis of human judgments and beliefs by integrating an experimental set-up in a survey [29]. They are, therefore, increasingly used to study hiring decisions [30–33]. In the context of the current hiring experiment, genuine recruiters evaluated fictitious applicants depicted in written resumes, for which the characteristics ('vignette dimensions', among which is the number of spelling errors in a resume) varied systematically over several categories ('vignette levels', e.g., zero, two or five errors). As vignette experiments combine experimental and survey elements, they inherit favourable attributes of both causal interpretations (experiment) and increased external validity through investigation of a broader (survey) population. Additionally, through the (i) manipulation of multiple applicant characteristics (besides spelling errors) and a (ii) randomisation across eight job contexts, we again ensure the external validity of our study. Finally, by eliciting applicant perceptions per hiring decision made, our study design allows us to measure explanations for different spelling error penalties inflicted upon applicants.

The main limitation of survey experimental designs is their hypothetical nature. More concretely, they measure behavioural intentions rather than actual behaviour. Nevertheless, intentions are a core proxy of behaviour [34]. Regarding the predictive validity of survey experiments, Hainmueller, Hangarter and Yamato [35] illustrate that survey experiments can predict real-world behaviours. Indeed, in their review Treischl and Wolbring [36] discuss the validity of survey experiments and suggest that it tends to be the highest when engaged respondents familiar with the decision process (here: hiring) complete tasks that remain faithful to the real-world context, for instance, by providing realistic vignette content.

A common alternative to study hiring processes are correspondence experiments where fictitious applications are sent to actual vacancies (see Lippens, Vermeiren and Baert [37] for an overview). However, whereas correspondence study actual hiring behaviours, they are limited in the sense that they do not allow researchers to survey the decision-maker–which is a crucial contribution of our current study. Applications of survey and correspondence experiments in the study of hiring discrimination are therefore complementary in nature. For instance, both vignette [38] and correspondence [37] experiments report evidence for ageism. However, whereas correspondence experiments provide estimates in a field setting, vignettes explain ageism through its driving perceptions.

### 2.1. Vignette design

In our experiment, participants passed a series of judgments on three fictitious resumes ('vignettes'). Applicants were all Flemish graduates. Flanders is the Dutch-speaking, northern part of Belgium and its largest community. More than 6.5 million people in the Belgian population (of 11.5 million people in total) live in Flanders. The Flemish labour market (Belgium) is mainly characterised by a (1) relatively high competition for human capital (compared to other regions in Europe) and a (2) high regulation of labour contracts [39]. The development of graduate resumes had the advantage that disproportionally dominant vignette dimensions such as relevant experience were avoided [29]. Concise resumes were realistic for this population; consequently, our manipulations were not overly complex for the different jobs. We return to this point in Section 4.

Fictitious applicants varied systematically across seven vignette dimensions on pre-determined levels [29]. Furthermore, our vignettes were based on resume templates of the Public Employment Agency of Flanders (PEAF). The exact vignette dimensions and their corresponding levels employed are discussed below and summarised in Table 2.

This first and most critical dimension in which fictitious job applicants differed was the number of spelling errors a resume contained, namely zero, two or five errors. Errors in resumes violated the rules of spelling by, for example, incorrectly conjugating verbs or incorrectly spelling words phonetically. Incorrect conjugations of verbs resulting from an erroneous implementation of the 'dt-rule' are the most common spelling errors in Dutch [40]. Other examples of implemented errors are: 'manlijk' (correct: 'mannelijk'; male (English)) and 'vollybal' (correct: volleybal; volleyball (English)). Cross-validation of errors was performed by two native Dutch-speaking researchers and twenty pretesters (i.e., pretesters (i) recognised the errors in resumes and (ii) evaluated the proposed errors as realistically occurring). The complete record Dutch spelling errors used in the experiment is displayed in Appendix Fig 1 in S1 File. Theoretically, word-processing software should be able to trace many errors we implemented in our experimental materials–and these errors could be avoided in practice. However, in line with our pretesters' evaluation of the errors as 'realistically occurring', applicants might not use word-processing software at all (e.g. when formatting a resume in online tools) or incorrectly (e.g. failing to adjust the settings when languages are combined in documents (e.g. Dutch and English)).

The following dimensions were manipulated in the resume heading: (2) applicant sex ('male', 'female') and (3) age of graduation ('foreseen age of graduation', 'one year past the foreseen age of graduation' and 'two years past the foreseen age of graduation').

The remaining manipulated dimensions were (4) student work experience, (5) hobbies, (6) achievement in tertiary education and (7) applicant's perception of their mother tongue. Following Van Belle and colleagues [33], we made the following distinctions in (4) student work: 'none mentioned', 'student work in the weekends' and 'student work during the holidays'. Hobby (5) categories included 'none mentioned', 'team sports' and 'volunteering', which were

**Table 2. Vignette dimensions and levels presented in experimental materials.**

| Vignette dimensions | Vignette levels |
|---|---|
| Number of spelling errors | {0; 2; 5} |
| Sex | {Male; Female} |
| Age of graduation (years)[a] | {Foreseen age of graduation; Foreseen age of graduation + 1; Foreseen age of graduation + 2} |
| Student work | {None mentioned; Student work in the weekends; Student work during the holidays} |
| Hobbies | {None mentioned; Team sports; Volunteering} |
| Achievement in tertiary education[b] | {None mentioned; Graduated cum laude; International experience} |
| Perception of mother tongue | {Mother tongue; Mother tongue, excellent} |

Notes. As described in Subsection 2.1, 99 applicant resumes (i.e., combinations of seven vignette dimensions filled out in resume templates) were systematically bundled in 33 decks of three graduate resumes. Participants were then randomly assigned one deck to evaluate

[a] In Flanders, the foreseen ages of graduation are 18 (secondary education) and 22 (tertiary education at the used master levels) years.

[b] The level of achievement in tertiary education was fixed to 'none mentioned' when participants were assigned to a vacancy with lower educational requirements (see Subsection 2.3).

based on multiple earlier hiring experiments [32, 33, 39]. Achievement in tertiary education (6) was distinguished by 'none mentioned', 'graduated cum laude' and 'international experience'. The variable was logically fixed at the value 'none mentioned' in resumes from high school graduates. Applicant's perception of their mother tongue (7) had two levels: 'mother tongue' and 'mother tongue, excellent'. The second level stressed graduates' varying self-rated language capacities. In practice, resume sections for known languages indeed allow applicants to express their perceived language mastery in different degrees.

The seven dimensions described were logical choices to manipulate in fictitious (graduate) resumes to maximise the external validity of our experiment as the dimensions are commonly presented in real-life resumes. Moreover, earlier research has evidenced the dimensions' relevance in recruiters' decision-making [4, 7, 33]. In particular for the Flemish hiring context, see the field experiments on the effect of gender, grade retention, volunteering and achievement in tertiary education on employment opportunities of Baert and colleagues [41], Baert and Picchio [42], Baert and Vujić [43] and Baert and Verhaest [39]. Note that manipulating fewer relevant dimensions could have led to an overestimation of the spelling-error penalty inflicted by recruiters because (mimicking) a real-life hiring decision also requires recruiters to combine different sources of information. Furthermore, applying a theoretical framework of signalling, interaction effects were expected between spelling errors and (other) applicant characteristics on hiring chances. In particular, (in)consistencies between signals [5] could impact the magnitude of spelling errors' effect on hiring chances. For instance, descriptive gender stereotypes such as 'women are more perceptive and understanding' [44] could cause a female gender to signal higher interpersonal skills to a recruiter. When spelling errors convey an opposite signal (i.e., lower interpersonal skills), an inconsistency arises between the two signals. This inconsistency might result in a buffering effect (i.e., women receiving a lower penalty) or a strengthening effect (i.e., women receiving a higher penalty).

Our selection of dimensions and levels resulted in a total of 972 possible unique vignette combinations (i.e., 3 (spelling errors) × 2 (sex) × 3 (age of graduation) × 3 (student work) × 3 (hobby) × 3 (tertiary education-specific achievement) × 2 (language perception of mother tongue)). Obviously, a fully factorial design in which every participant rated each unique vignette would have put unreasonable demands on the participants (and having each participant judge a single vignette would have been inefficient for data collection). Hence, we constructed a D-efficient design [29], selecting subsets of vignettes ('decks') with limited correlations between dimensions and being balanced across levels. More concretely, the algorithms stacked a subset of 99 unique vignettes into 33 decks of three vignettes. The resulting D-efficiency score of 98.347 (maximum 100) indicated that any losses of estimation precision were negligible compared to a fully factorial design in which all 972 vignettes were judged by participants.

In addition, the fictitious resumes presented in the experiment comprised (i) a typical Flemish sounding name, (ii) a postal address in a middle-class neighbourhood, (iii) a mobile phone number, (iv) an e-mail address with a major provider, (v) a date of birth, (vi) an indication of the Belgian nationality and (vii) the most logical educational degree for the job. The layout of the resumes also differed slightly –as mentioned above, templates from the Public Employment Agency of Flanders were used. These small differences cannot bias our research results since the aforementioned seven vignette factors were randomised across the templates. An example vignette is presented in Appendix Fig 2 in S1 File.

## 2.2. Data collection

In April 2020, our vignette experiment was integrated into a large-scale online survey wave for different research projects. In total, 38,432 invitations were sent to genuine HR-professionals

living in Flanders. Panel members received an invitation and two weeks later they were sent a single reminder. Hence, a total of 4,084 professionals started one of the wave's surveys (global response rate: 10.6%). However, only the participants indicating hiring experience qualified for our study. Consequently, 1,026 surveys were started for the current study. Next, participants were withheld in our final study sample when they (i) responded affirmatively to a question on recent hiring experience ('In the past year, were you responsible for recruiting candidates (applicants)'), (ii) completely filled out the survey and (iii) passed a manipulation check (i.e., responding affirmatively to the question 'Did the applicants you judge differ in language care?' near the end of the experiment). After applying these criteria, we withheld 445 professionals and 1,335 vignette observations (selection rate: (445/1,026)*100 = 43.4%).

## 2.3. Procedure

**Job vacancy.**   Following a written informed consent form, participants received detailed experimental instructions. Participants had to imagine themselves as the head of human resources in the company 'Peeters NV' (a neutral-sounding Flemish family name) with the current task to fill a specific vacancy.

Then, one out of eight fictitious job vacancies was randomly assigned to the experimental recruiters. Adding to the external validity of the experiment, we pursued variation in three job characteristics. First, the presented vacancies varied in the required level of education. Second, the presented vacancies varied with regard to the type of sector. Applying the macro-economic distinction between the secondary and tertiary sectors, we presented both manufacturing and services vacancies. By manipulating both the required education level and sector in the hiring assignment, our study's set-up is distinguished from earlier work that exclusively investigated the role of spelling errors in service-sector occupations with high educational requirements (so-called 'white collar' jobs; for an overview, see Appendix Table 1 in S1 File). Third, vacancies varied with regard to their required written communication skills. In line with Ehrhart and Ziegert [45], we expect the 'signal receivers', here recruiters, to apply a different weight to the signal of spelling errors when hiring for jobs with higher written communication requirements because writing skills have a more immediate relevance when hiring for said jobs [27].

To systematically operationalise variations in the proposed job characteristics, we based ourselves on the O*Net classifications of occupations. More concretely, we approximated each experimental job characteristic (discussed supra) with its corresponding O*Net attribute(s) and then selected occupations according to their O*Net scores per attribute. More specifically, we matched required education level to its direct O*Net counterpart. Sector was inferred based on the task descriptions of O*Net occupations. Required written communication skills was a weighted average of the attributes 'Communicating with Persons Outside Organization', 'Communicating with Supervisors, Peers or Subordinates', 'Electronic Mail', 'Writing' and 'English Language'. As a result of this exercise, participants received a fictitious vacancy for one of the following occupations: (i) recreation worker, (ii) production worker, (iii) secretary, (iv) assistant graphic designer, (v) specialist electronics, (vi) air traffic controller, (vii) human resources manager and (viii) audiovisual specialist. The vacancy descriptions presented to participants were also derived from their respective O*Net descriptions. An overview of the job characteristics per corresponding occupation is presented in Appendix Table 1 in S1 File.

**Resume evaluations.Hiring decisions.**   After reading through their assigned vacancy, participants judged three graduate resumes –varying on the seven dimensions discussed in Subsection 2.1 –by sharing their hiring decisions and perceptions of applicants on response scales from 0 (fully disagree) to 10 (fully agree). An overview of items used for the vignette evaluations is shown in Table 3 below.

**Table 3. Items used for applicant evaluations.**

| Evaluative dimension | Statement |
|---|---|
| A. HIRING DECISION | |
| Interview probability | I think that I will invite this applicant for a job interview. |
| Hiring probability | There is a high chance that I will effectively hire this applicant. |
| B. PERCEIVED INTERPERSONAL SKILLS | |
| Perceived quality of communication | I think that this applicant will communicate well with me |
| Perceived quality of communication during a job interview | I think that this applicant will communicate well with me during a job interview. |
| Perceived ability to get along with others encountered on the job | I think that the applicant will get along with all sorts of people she/he will encounter in this job. |
| Perceived pleasure in interaction | I think that, at work, I will enjoy interacting with this person. |
| C. PERCEIVED CONSCIENTIOUSNESS | |
| Perceived as hard-working | I think that this person is hard-working. |
| Perceived as organised | I think that this person will work in an organised manner. |
| Perceived as thorough | I think that person will work thoroughly. |
| Perceived as systematic | I think that this person will work systematically. |
| Perceived as being responsible | I think that this person is responsible. |
| D. PERCEIVED MENTAL ABILITIES | |
| Perceived problem-solving ability | I think that this person has strong problem-solving abilities. |
| Perceived capacity to learn quickly | I think that this person has the capacities to quickly learn new skills. |
| Perceived intelligence | I think that this person is intelligent. |
| Perceived knowledgeability | I think that this person is knowledgeable. |

Notes. Each item was rated on a scale from 0 (Completely disagree) to 10 (Completely agree).

More concretely, recruiters' decisions were measured by two statements. Responding to a first statement, participants rated the probability they would invite an applicant for a first job interview (hereafter referred to as 'interview probability'). This is our benchmark outcome variable. Subsequently, participants rated the probability they would eventually hire the applicant using a second statement ('hiring probability'). This outcome is used in our robustness analysis.

**Resume evaluations. Applicant perceptions.** Immediately following their hiring decisions, participants shared their applicant perceptions using 13 statements as derived from the literature on spelling errors' potential signalling effects (section 1, supra). In total, three perception scales were calculated by adding up the corresponding perception items (infra) and dividing them through the number of items of that scale. Consequently, their resulting scores also range from 0 to 10. As a robustness check, analyses were separately conducted for scale and item level.

First, as a measure of *perceived interpersonal skills*, we translated Finkelstein and Burke's [46] interpersonal skill scale into Dutch, which is a scale developed with assessing managers' perceptions of applicants in mind. Furthermore, after revision of the scale by experts in the field of labour market research, we complemented the scale with a fourth item on the general quality of communication with the applicant beyond job interviews. Hence, items measured the applicant's expected (i) quality of communication in general, (ii) communication during a job interview, (iii) ability to get along with others encountered on the job as well as (iv) the recruiter's perceived pleasure of interacting with the applicant.

Second, *perceived conscientiousness* was measured with five items drawn from a Dutch translation of Costa and McCrae's [47] seminal work. More specifically, items gauged perceptions of applicants regarding whether they work (i) hard, (ii) in an organised manner, (iii) thoroughly, (iv) systematically and (v) were responsible.

Third, *perceived mental abilities* were assessed through a scale of four items based on the seminal works of Dunn, Mount, Barrick and Ones [48] and Warner and Sugarman [49]. That is, items gauged perceptions of applicants' (i) problem-solving abilities, (ii) capacity to quickly learn new skills, (iii) intelligence and (iv) knowledgeability.

**Post-experimental questionnaire.** Upon having indicated their evaluations of three fictitious resumes, recruiters completed the experimental procedure by filling out a post-survey questionnaire with participant variables. These variables included demographic characteristics as well as professional experience and psychographic variables. The data were to be used in moderation analyses and robustness checks.

More concretely, the demographics surveyed were: participant gender ('male', 'female'), age (in years, continuous), mother tongue ('Dutch', 'other'), nationality ('Belgian', 'other EU-28', 'other non-EU-28') and education level ('tertiary education at university', 'tertiary education outside of university', 'secondary education', 'primary education'). Participants' professional experiences were gauged through an item on hiring tenure ('less than a year', 'one to five years', 'greater than five years').

The two psychographic variables investigated were sensitivity to language care and social desirability. Whereas Martin-Lacroux and Lacroux [27] found that recruiters' penalties inflicted for spelling errors were moderated by their own spelling capacities, we complement their work by testing whether participants' self-reported language sensitivity has a similar moderating role. Indeed, based on signalling theory, recruiters ('signal receivers') who report being sensitive to the language care of job applicants might assign additional weight to spelling errors in resumes ('receiver calibration' [5]). Sensitivity to language care was measured with a self-developed scale consisting of three statements: (i) 'In general, language care is important to me', (ii) 'When judging job applicants, language is important to me' and (iii) 'I am language-sensitive'. These statements were rated on a scale from 0 'fully agree' to 10 'fully disagree', summed and scaled to 10 (Cronbach's $\alpha = 0.897$). Finally, participants' social desirability was measured through the shortened Marlowe-Crowne Social Desirability Scale developed by Reynolds [50] and validated across different contexts [51, 52]. The scale contains 13 items expressing behaviours that are either socially sanctioned or approved (e.g., 'I sometimes feel resentful when I don't get my way'). Participants indicated whether the items applied to them (score 1) or not (score 0). Afterwards, participants' total social desirability scores were calculated by summing item scores.

## 2.4. Data description

In our sample, both sexes were well-represented (46.7% women). On average, participants were 47.011 years old ($SD = 11.165$), had Dutch as their mother tongue (96.6%), enjoyed tertiary education (91.5%) and tended to agree with statements on their sensitivity to language care –with an average score of 7.968 out of 10 ($SD = 1.484$). The samples' eligibility was further indicated by participants' professional background because about half (47.9%) of the respondents had more than five years of experience making hiring decisions.

Furthermore, the low- to non-existent correlations between spelling errors and other applicant (minimum: 0.017, maximum 0.080), job (minimum −0.001, maximum: 0.030) and participant (minimum: 0.001, maximum 0.018) characteristics indicated our D-efficient design (Subsection 2.1) and randomisations were successful.

## 2.5. Statistical framework

The experimental data collected for this study were analysed with the 'Stata/MP 15' statistical software using regression methods. To estimate the total effects of two and five spelling errors, we first linearly regressed the hiring outcomes on the different applicant, job and participant characteristics (Subsection 2.3), with standard errors clustered at the participant level. Logistic regressions did not change our research conclusions. Second, we investigated heterogeneity in the penalties inflicted for two and five errors by testing moderation effects related to the afore-mentioned applicant, job and participant characteristics. Third, as shown in Fig 1 below, we applied a multiple mediation framework [53] to decompose the total effects of spelling errors (c-path) on hiring decisions by potential underlying mechanisms. This model is explained in Subsection 3.3.

# 3. Results

## 3.1. The effects of spelling errors on interview probability

In line with the existing literature discussed in Section 1, we find that recruiters inflict hiring penalties for spelling errors in resumes. Indeed, as shown in Table 4's column (1A), resumes featuring two ($\beta = -0.730$, p < 0.001) and five ($\beta = -1.850$, p < 0.001) errors receive lower interview probabilities. More specifically, the graduate resumes containing five spelling errors receive an 18.5 percent points lower interview probability compared to an error-free resume. When we compare our study's subsample estimate for five spelling errors in the context white-collar occupations (−12.2 percent points), we find that it is well comparable to a study among HR recruiters by van Toorenburg and colleagues [26] reporting a 13.7 percent points decrease in hireability ratings of HR specialists for resumes featuring five spelling errors.

Furthermore, having manipulated additional applicant characteristics, we can compare the magnitudes of spelling errors' effects with those of other resume characteristics. Hence, for our graduate resumes, we find that the two-error penalty is comparable to the value recruiters attach to applicants' volunteering experience (compared to no extracurricular activity men-tioned; $\beta = -0.706$, $p < 0.001$; Table 4, column (1A)). The impact of five spelling errors, how-ever, is substantially greater than that of any other experimental manipulation. Therefore, no 'counterpart' is evident among the manipulations.

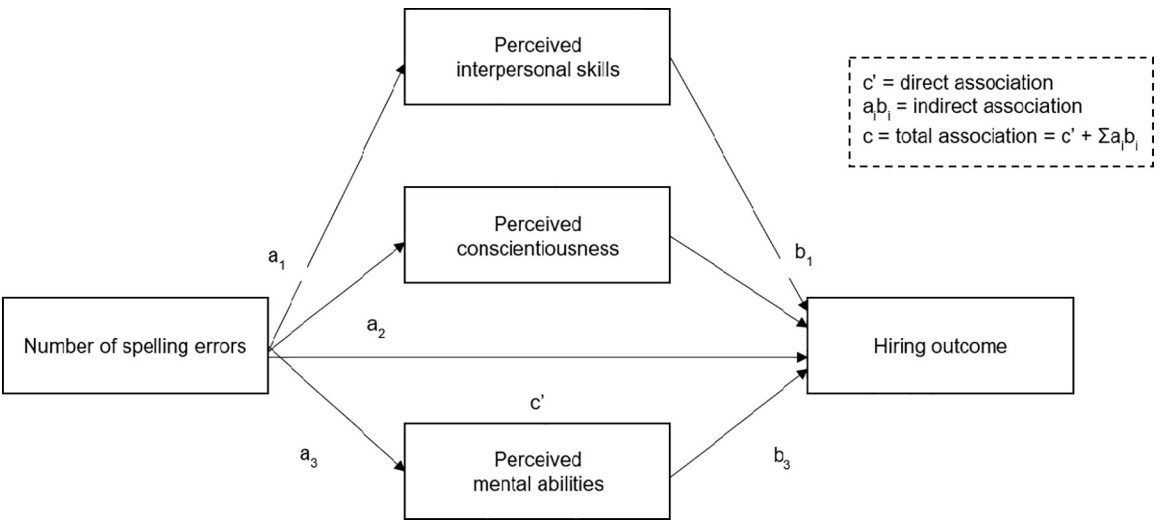

**Fig 1. Core mediation framework outlined in Subsection 2.5.**

**Table 4. Regression results with interview probability as the outcome variable, two-way interactions included for five spelling errors.**

| | Interview probability | | | | |
|---|---|---|---|---|---|
| | **(1)** | **(2)** | **(3)** | **(4)** | **(5)** |
| A. APPLICANT CHARACTERISTICS | | | | | |
| Spelling errors (ref. = none) | | | | | |
| Two errors | −0.730*** (0.135) | −0.719*** (0.136) | −0.734*** (0.136) | −0.730*** (0.135) | −0.726*** (0.136) |
| Five errors | −1.850*** (0.155) | −2.388*** (0.584) | −1.671*** (0.218) | −0.127 (0.880) | −0.349 (1.070) |
| Female | 0.375*** (0.121) | 0.620*** (0.173) | 0.394*** (0.119) | 0.368*** (0.120) | 0.631*** (0.171) |
| Age of graduation (ref. = foreseen age) | | | | | |
| One year later | 0.164 (0.146) | −0.124 (0.207) | 0.120 (0.145) | 0.167 (0.145) | −0.149 (0.205) |
| Two years later | 0.137 (0.143) | −0.133 (0.200) | 0.100 (0.142) | 0.159 (0.141) | −0.138 (0.198) |
| Student work (ref. = none mentioned) | | | | | |
| On the weekends | 0.308** (0.145) | 0.323 (0.204) | 0.345** (0.146) | 0.312** (0.145) | 0.343* (0.205) |
| During holidays | 0.227* (0.132) | 0.109 (0.204) | 0.230* (0.132) | 0.233* (0.132) | 0.095 (0.205) |
| Hobbies (ref. = none mentioned) | | | | | |
| Team sports | 0.308** (0.141) | 0.260 (0.208) | 0.283** (0.140) | 0.309** (0.142) | 0.236 (0.209) |
| Volunteering | 0.706*** (0.139) | 0.414** (0.179) | 0.678*** (0.139) | 0.700*** (0.138) | 0.390** (0.178) |
| Achievement in tert. edu. (ref. = none mentioned) | | | | | |
| Graduated cum laude | 0.124 (0.214) | 0.175 (0.247) | 0.133 (0.215) | 0.136 (0.214) | 0.391 (0.266) |
| International experience | 0.438** (0.213) | 0.273 (0.262) | 0.414* (0.211) | 0.455** (0.211) | 0.514* (0.284) |
| Mother tongue perceived as excellent | 0.155 (0.128) | 0.278 (0.173) | 0.166 (0.127) | 0.138 (0.127) | 0.262 (0.173) |
| Five errors × Female | | −0.718* (0.362) | | | −0.691* (0.366) |
| Five errors × Graduated one year later | | 0.718 (0.449) | | | 0.674 (0.454) |
| Five errors × Graduated two years later | | 0.676 (0.451) | | | 0.665 (0.452) |
| Five errors × Student work on the weekends | | −0.078 (0.441) | | | −0.067 (0.444) |
| Five errors × Student work during holidays | | 0.368 (0.442) | | | 0.359 (0.442) |
| Five errors × Team sports | | 0.118 (0.452) | | | 0.132 (0.448) |
| Five errors × Volunteering | | 0.888** (0.404) | | | 0.828** (0.405) |
| Five errors × Graduated cum laude (tert. edu.) | | 0.014 (0.479) | | | −0.656 (0.602) |
| Five errors × International experience (tert. edu.) | | 0.634 (0.448) | | | −0.032 (0.565) |
| Five errors × Mother tongue perceived as excellent | | −0.221 (0.357) | | | −0.248 (0.359) |
| B. JOB REQUIREMENTS | | | | | |
| Req. level of education: high | 0.040 (0.274) | −0.054 (0.224) | −0.240 (0.278) | 0.036 (0.275) | −0.382 (0.289) |
| Req. written communication: high | −0.649*** (0.222) | −0.581*** (0.224) | −0.464** (0.230) | −0.646*** (0.223) | −0.407* (0.233) |
| Req. type of labour: service | 0.023 (0.216) | 0.032 (0.215) | 0.154 (0.227) | 0.030 (0.216) | 0.167 (0.226) |
| Five errors × high level of education req. | | | 0.850*** (0.278) | | 1.017** (0.399) |
| Five errors × high written communication req. | | | −0.558** (0.262) | | −0.509* (0.268) |
| Five errors × service labour req. | | | −0.407 (0.276) | | −0.395 (0.285) |
| C. PARTICIPANT CHARACTERISTICS | | | | | |
| Female | 0.139 (0.223) | 0.133 (0.222) | 0.138 (0.223) | 0.100 (0.232) | 0.057 (0.232) |
| Age (c) | −0.001 (0.009) | 0.001 (0.009) | −0.001 (0.009) | 0.012 (0.012) | −0.004 (0.010) |
| Language sensitivity (c) | −0.110 (0.068) | −0.113 (0.069) | −0.109 (0.068) | −0.011 (0.072) | −0.010 (0.072) |
| Five errors × Female | | | | 0.095 (0.267) | 0.187 (0.262) |
| Five errors × Age | | | | 0.012 (0.012) | 0.013 (0.012) |
| Five errors × Language sensitivity | | | | −0.294*** (0.082) | −0.303*** (0.082) |
| N | 1,335 | | | | |

Notes. Abbreviations used: c (continuous variable), ref. (reference category), req. (required), and tert. edu. (tertiary education). The presented statistics are coefficient estimates and their standard errors in parentheses. Standard errors are corrected for clustering of the observations at the participant level.

***, ** and * indicate significance at the 1%, 5%, and 10% levels, respectively.

## 3.2. Different moderators of the spelling error penalty

To further our understanding of the interview penalties inflicted for spelling errors left in resumes, we now employ moderation analyses to examine whether the effects of two and five spelling errors are attenuated or amplified in co-occurrence with other (i) applicant, (ii) job and (iii) participant characteristics (Section 1). As participants could vary in unobserved, confounding variables, a causal interpretation of the interactions regarding the recruiter characteristics is inappropriate. Our set-up does, however, provide causal evidence for moderation effects between the different applicant characteristics as well as those related to job characteristics, as both were carefully manipulated.

Panels (2) to (5) of Table 4 contain the results of a stepwise insertion of interaction terms with applicant (panel A), job (panel B) and participant (panel C) characteristics. For conciseness, we limit ourselves to a presentation of the analyses with five errors in the main text. Nonetheless, the complete analyses with two errors are displayed in Appendix Table 2 in S1 File. Our analyses of the two-way interaction terms with two spelling errors suggest that the penalty inflicted for two errors is uniform across the different applicant, job and participant characteristics investigated (Section 2) as we cannot identify statistically significant interaction effects. Regarding the job characteristics, however, we find it remarkable that the inflicted penalties for two errors are similar for occupations with higher written communication requirements ($\beta$ = 0.050, $p$ = 0.847; Appendix Table 2 in S1 File, column (5B)). This is surprising given that, in some sense, reading through applicants' resumes could already be considered a preliminary 'work sample' for those jobs in which written communication takes a central role [54].

Interestingly, the analyses of interaction effects with five spelling errors yield a different picture. Our data suggest substantial heterogeneity in the penalty for five errors across applicant, job and participant characteristics. For instance, we find that female applicants applying with an error-laden resume are penalised more severely than males, albeit with marginal significance ($\beta$ = −.691, $p$ = 0.060; Table 4, column (5A)). As indicated in Section 1, this effect could result from a specific case of signal inconsistency, namely 'gender incongruent behaviour' [55]. Specifically, recruiters could interpret spelling errors in a resume as a violation of behavioural norms for women, but less so for men. In Subsection 3.3 below, we briefly return to this point by providing evidence for differences in recruiters' expectations ('signals') between male and female applicants. Furthermore, we find evidence for a buffering effect for applicants who mention volunteering as an extracurricular activity. Volunteers receive a lower penalty for five spelling errors featured in their resume ($\beta$ = 0.828, $p$ = 0.041; Table 4, column (5A)). One explanation for this buffering effect could be that recruiters perceive volunteering as a reliable signal [5] of interpersonal skills [56] in comparison to spelling errors. Therefore, recruiters may resolve this signalling inconsistency by inflicting a lower penalty for errors.

In our analyses with job characteristics, we similarly identify a buffering effect. When applying for jobs with high educational requirements (white-collar occupations in our experiment), more concretely, one is penalised less severely for an error-laden resume than when applying for jobs with lower educational requirements ($\beta$ = 1.017, $p$ = 0.011; Table 4, column (5B)). An explanation for this effect could be similar to that of volunteering (supra). As such, applicants' level of education, like spelling errors, already has a reliable signalling function for intelligence [56]. Hence, higher or lower levels of education could attenuate or amplify, respectively, the penalty inflicted for errors, which is a result of signalling inconsistency. Next, in contrast to the moderation analyses for two errors, there is a marginally significant interaction with the occupation's written communication requirements. More specifically, we find that in the case of jobs with high written communication requirements, five spelling errors are penalised more severely ($\beta$ = −0.509, $p$ = 0.058; Table 4, column (5B)).

We conclude our moderation analyses for five spelling errors with an analysis of participant characteristics. In contrast to resumes with two errors, we find that higher scores of recruiters' sensitivity to language care (Subsection 2.3) are associated with more severe penalties inflicted for five spelling errors ($\beta = -0.303$, $p < 0.001$; Table 4, column (5C)). As introduced in Section 1, once again applying signalling theory, this interaction could be understood as a case of 'receiver calibration' where more sensitive recruiters apply higher weights to language care when screening error-laden resumes.

### 3.3. Testing spelling errors' signalling functions as an explanation

Next, we investigate the underlying mechanisms for the penalties found related to the perceptions of the candidates' interpersonal skills, conscientiousness and mental abilities. That is, we investigate (i) to which extent spelling errors in the candidates' resumes affect these perceptions and (ii) to which extent these perceptions explain variation in the interview probability.

To this end, we estimate of multiple mediation model [5]. For Fig 1, we calculate mediation effects (a × b-paths) and a remaining direct effect (c'-path). The experiment allows for a causal interpretation of both spelling errors' signals (a-paths) and total effects (c-path; discussed in Subsection 3.1). It is, however, important to acknowledge that the experiment cannot provide causal evidence for the associations between applicant perceptions and the interview invitation outcome (b-paths), for reasons similar to those discussed in Subsection 3.2.

In econometric terms, we estimate the following equations within our model:

$$M_1 = \alpha_{M_1} + \beta_{M_1}AC + \gamma_{M_1}JC + \delta_{M_1}RC + \varepsilon_1 TSE + \vartheta_1 FSE + \epsilon_{M_1};\tag{1}$$

$$M_2 = \alpha_{M_2} + \beta_{M_2}AC + \gamma_{M_2}JC + \delta_{M_2}RC + \varepsilon_2 TSE + \vartheta_2 FSE + \epsilon_{M_2};\tag{2}$$

$$M_3 = \alpha_{M_3} + \beta_{M_3}AC + \gamma_{M_3}JC + \delta_{M_3}RC + \varepsilon_3 TSE + \vartheta_3 FSE + \epsilon_{M_3};\tag{3}$$

$$Y = \alpha_Y + \beta_Y AC + \gamma_Y JC + \delta_Y RC + \varepsilon' TSE + \vartheta' FSE + \theta_1 M_1 + \theta_2 M_2 + \theta_3 M_3 + \epsilon_Y.\tag{4}$$

In this model, $M_1$, $M_2$ and $M_3$ are mediation scales: perceived interpersonal skills, perceived conscientiousness and perceived intelligence, respectively (or the 13 underlying items, in our secondary analysis). *TSE* and *FSE* represent applicants with two and five spelling errors in their resumes, respectively. *AC* is a vector of the additional applicant characteristics (Subsection 2.1), *JC* a vector of the three job characteristics (Subsection 2.3) and *RC* a vector of the measured participant characteristics (Subsection 2.3). *Y* is the interview (or hiring) probability. Then, $\beta_M$, $\gamma_M$, $\delta_M$, $\varepsilon_i$ and $\vartheta_i$ are the parameters associated with *AC*, *JC*, *RC*, *TSE* and *FSE* in the equations with $M_i$ as dependent variable and $\alpha_{M_i}$ as the intercept. $\beta_Y$, $\gamma_Y$, $\delta_Y$, $\varepsilon'$, $\vartheta'$ and $\alpha_Y$ are the corresponding parameters in the fourth equation with *Y* as dependent variable. Last, $\theta_1$, $\theta_2$ and $\theta_3$ are parameters associated with the mediator scales in the fourth equation.

From these equations $\varepsilon'$ and $\vartheta'$ are the remaining direct effects of two and five spelling errors after controlling for the mediators. In this study, we are mainly interested in the parameters (i) $\varepsilon_i$, $\vartheta_i$ and (ii) the products $\varepsilon_i\theta_i$, $\vartheta_i$, $\theta_i$. These represent the (i) signalling effects of spelling errors and the (ii) indirect effects of spelling errors on *Y* through the three mediators. Following Hayes [53], we estimate the four equations simultaneously and correct standard errors (i.e., $\epsilon_{M_1}$, $\epsilon_{M_2}$, $\epsilon_{M_3}$ and $\epsilon_Y$) for clustering observations at the recruiter level. Appendix Table 3 in S1 File presents the full estimation results.

As theorised, both two and five spelling errors convey negative signals to employers. More concretely, we find causal evidence that spelling errors instil perceptions of lower (i)

**Table 5. Mediation analysis: Percentages of spelling error's effects on interview probability explained by each mediating scale.**

| Mediators | Two spelling errors | Five spelling errors |
|---|---|---|
|  | Percentage of spelling error's effect on interview probability explained by mediators [p-value] | Percentage of spelling error's effect on interview probability explained by mediators [p-value] |
| Perceived interpersonal skills (s) | **9.0%** [0.038] | **11.6%** [0.033] |
| Perceived conscientiousness (s) | **12.1%** [0.049] | **11.9%** [0.001] |
| Perceived mental abilities (s) | **32.2%** [0.000] | **31.9%** [0.000] |
| N | 1,335 | |

Notes. Abbreviation used: s (scale consisting of multiple items). P-values are corrected for clustering of observations at the participant level. Percentages related to p-values below 5% are in bold.

interpersonal skills (two errors: $\beta = -0.228$; $p = 0.006$; five errors: $\beta = -0.745$; $p < 0.001$), (ii) conscientiousness (two errors: $\beta = -0.427$; $p < 0.001$; five errors: $\beta = -1.072$; $p < 0.001$) and (iii) mental abilities (two errors: $\beta = -0.399$; $p < 0.001$; five errors: $\beta = -1.000$; $p < 0.001$). In that the response scales range from 0 to 10, we can interpret the coefficients as follows: five spelling errors lead to a 10.0 percent points reduction in perceived mental abilities of the applicant. Comparing the coefficients of the signalling effects, our model suggests that of the three scales, (i) the magnitudes of signalling effects are highest for perceived conscientiousness and perceived mental abilities and (ii) five errors, compared to two, elicit the same yet 'stronger' perceptions from recruiters.

Furthermore, returning to the applicant-side moderators from Subsection 3.2, our rich experimental set-up also allows for the estimation of signalling effects of other resume elements –an advantage it has over earlier studies mentioned in Table 1. For instance, we find that volunteering signals higher interpersonal skills ($\beta = -0.711$; $p < 0.001$), corroborating the explanations we propose for the moderating role of applicants' volunteering. Likewise, we find that recruiters expect women to possess higher levels of interpersonal skills, conscientiousness and mental abilities than men (see Appendix Table 3 in S1 File for a complete overview of effects).

Next, although we find evidence for multiple signals emitted by spelling errors, not all signals necessarily predict recruiters' interview decisions to the same extent. Investigating the signals' potential as mediators, we calculate the indirect effects of two and five spelling errors on interview probability via the proposed mediators over a bootstrapping procedure. Table 5 presents the percentage of the total spelling error effect on the interview outcome explained by each mediator. For the resumes featuring two errors, we find that perceived interpersonal skills (9.0% of the total effect), perceived conscientiousness (12.1%) and perceived mental abilities (32.2%) explain significant shares of the interview penalty. Again, we identify a similar pattern for resumes with five errors.

Our conclusion that perceived mental abilities explains the largest share of the penalty– even though its signalling impact is comparable to that of perceived conscientiousness –can be understood by differences in recruiters' preferences. More concretely, our data suggest that recruiters value applicants' mental abilities over conscientiousness. From the indirect effects calculated, we conclude that our model partially mediates the effects of two (9.0% + 12.1% + 32.2% = 53.3%) and five (11.6% + 11.9% + 31.9% = 55.4%) spelling errors on interview probability. Another reason to label the mediation as partial is that the direct effects of two errors ($\beta = -0.342$; $p = 0.002$) and five errors ($\beta = -0.825$; $p < 0.001$) on the interview probability remain

significant (Appendix Table 3 in S1 File). The partial mediation suggests that with the three categories of signals we investigated, we can already explain more than half of the penalty inflicted. However, additional perceptions could be in play beyond those we investigated.

We checked the robustness of the results by using alternative econometric specifications and re-estimating our benchmark models for subsets of the data. In particular, we (i) swapped the interview probability scale for the hiring probability scale and conducted analyses on (ii) participants with low to average social desirability scores and (iii) those who had at least one year of experience in hiring decisions. Participants were considered as scoring 'high' on the social desirability scale when they had a score above the sample average plus one standard deviation (0.661 + 0.170 = 0.831). See Appendix Table 4 in S1 File for the estimation results. Out of these checks, we conclude that the findings remain robust under these adaptations. However, in the subsamples investigated, the statistical significance of the mediators related to interpersonal skills and conscientiousness are less significant (sometimes only at the 10% level).

Finally, Appendix Tables 5 and 6 in S1 File display a secondary analysis where we re-estimate our multiple mediation model after replacing the three mediation scales for the thirteen individual perception items. This detailing provides a more fine-grained analysis of the driving perceptions. With respect to interpersonal skills, the experimental recruiters fear above all poorer communication during a job interview and later in the workplace. When job candidates make five mistakes, recruiters also think that others will be less likely to cooperate with these candidates and that they will enjoy that cooperation less. Regarding the perception of lesser conscientiousness, those who make spelling mistakes are rated as less (i) hard-working, (ii) well-organised, (iii) thorough, (iv) systematic, and (v) responsible. Finally, regarding the perception of lesser cognitive qualities, those who make spelling mistakes are perceived as (i) having lower problem-solving ability, (ii) less trainable, (iii) less intelligent, and (iv) less knowledgeable. In other words, we find empirical evidence for all spelling error signals included in our experiment. We find that the individual perception items on communication during a job interview (perceived interpersonal skills), working thoroughly (perceived conscientiousness) and knowledgeability (perceived mental abilities) were the most outspoken drivers of each scales' indirect effects.

## 4. Conclusion

To understand how spelling errors in job candidates' resumes drive their hiring chances, we conducted a scenario experiment in which genuine recruiters evaluated fictitious applicants with resumes containing a randomised numbers of spelling errors. More concretely, they evaluated three resumes for one out of eight job vacancies with respect to hireability as well as to 13 statements derived from the dominant theoretical perceptions about applicants associated with poor language care in resumes. Broadly speaking, our study makes four contributions to the literature. First, we drastically enhanced the ecological validity of scenario experiments on language care by allowing lower numbers of spelling errors in resumes. Indeed, as indicated in the literature overview from Table 1, prior experiments consistently featured five or more errors in a single resume. Second, to our knowledge, this work was the first to investigate the effects of spelling errors in blue-collar (besides white-collar) occupations. Third, through our design, we were able to realistically analyse the effects of spelling errors relative to other applicant characteristics featured in resumes and investigate whether these characteristics (as well as recruiter characteristics) moderate the spelling error penalty. Last, we quantitatively broke down hiring penalties into their underlying perceptions of applicants thereby uncovering the mechanisms underlying the spelling error penalty more comprehensively than previous studies.

In line with prior research, we conclude that graduates with an error-laden resume (featuring five errors) have an 18.5 percentage points reduced chance of an interview than applicants with an error-free resume. This magnitude of five errors' effect is unlike any of the other applicant-side manipulations we investigated (e.g., student work and hobbies). Moreover, we also find causal evidence for a similar, yet lesser, penalty inflicted to applicants with a smaller number of errors left in their resumes. That is, similar in magnitude to the hiring advantage caused by volunteering, we calculate that resumes with two errors receive 7.3 percent points lower interview probabilities.

Next, through our moderation analyses, we establish that there is substantial heterogeneity in the penalty inflicted for error-laden resumes –but less so for resumes with few errors. In particular, our data evidence that female applicants were penalised more severely than males and that the mention of volunteering had a buffering effect, thus reducing the penalty inflicted. Additional moderation analyses with job and participant characteristics suggest that error-laden resumes are more disapproved of in blue-collar jobs compared to white-collar jobs, in positions with high requirements for written communication and by recruiters perceiving themselves as sensitive to language-care.

Finally, we conclude that both two and five spelling errors have a negative signalling function for applicants' interpersonal skills, conscientiousness and mental abilities. Together, these three signals explain more than 50% of the interview penalties inflicted on resumes with two and five errors –with unfavourable perceptions of applicants' mental abilities explaining the largest share (approximately 30%). Similar to the hiring penalties inflicted, error-laden resumes also have a stronger impact on recruiters' perceptions of applicants than resumes with a lower number of errors.

The most obvious implication of this study's results is that applicants-to-be should carefully scan their applications for spelling errors as these prove to be costly mistakes in the hiring process. Indeed, recruiters disapprove of not only error-laden resumes but also, as we now evidenced, apply penalties for resumes containing relatively fewer errors.

Furthermore, from a scientist-practitioner perspective, our results raise concerns about the interrater reliability of resume screening. More specifically, as suggested by moderation analyses, recruiters' self-reported language sensitivity is associated with differential treatment of applicants who leave spelling errors in resumes. Perhaps of even larger concern is our finding that women (similar to ethnic minorities [28]) are penalised more severely for error-laden resumes. Therefore, to reduce inter-rater variability, optimise procedural fairness of hiring procedures [57] and grow a professional environment providing equal opportunities, organisations could consider organisation-wide guidelines on the application elements that are relevant for hiring practices.

Finally, we acknowledge this study's limitations and offer directions for future research avenues. First, the experimental design of the study allows us to causally infer (i) effects of spelling errors on both the hiring outcomes and applicant perceptions and (ii) the moderating role of other applicant and job characteristics. However, the associations between hiring outcomes, applicant perceptions and participant characteristics might correlate with exogenous confounding variables. Consequently, future researchers might want to investigate the potential causal effects of moderators at the recruiter side. Another avenue for future research could be to investigate different types of language errors. Indeed, whereas our design featured common spelling errors such as incorrect verb conjugations and improper phonetic spelling, we did not investigate complex grammatical errors. Experimental manipulations of cover letters that accompany (fictious) resumes could provide an ecologically valid context to study the impact of grammatical errors on job market prospects.

Second, following Charney and Rayman [14] and Charney and colleagues [25] we limited ourselves to manipulating realistic graduate resumes. By focussing on graduates, it was feasible

to richly manipulate additional resume characteristics in addition to the number of spelling errors without creating overly complex and dissimilar stimuli across different occupations. Indeed, an analogous design with more experienced applicants per job –eight in total –would have required supplementary and personalised information on work experiences and trajectories to become even remotely ecologically valid.

Third, the experiments' laboratory settings could have induced measurement biases because participants were aware of study participation. We mitigated this risk in the developmental phase of the study by developing multidimensional resumes that closely resemble real-life graduate applications. Indeed, vignette experiments are found to correlate strongly with actual behaviour [35]. We furthermore mitigated risks of measurement biases by performing robustness checks, for instance, on subsamples of participants with low to average social desirability tendencies. Nevertheless, follow-up research could eliminate the potential risks of lab experiments by developing a field experiment (e.g., a correspondence study [37]) to estimate interview penalties inflicted for different numbers and types of spelling errors featured in applications. Of course, this method comes at the cost of an inability to measure the underlying perceptions of applicants that recruiters have.

## Supporting information

**S1 File.**
(DOCX)

## Acknowledgments

We would like express our gratitude to Leen Pollefliet for her heartfelt display of enthusiasm and shared reflections on an earlier version of this manuscript. Data processing is organised in correspondence with Ghent University's GDPR-guidelines.

## Author Contributions

**Conceptualization:** Philippe Sterkens, Victor Van Driessche, Michael Geamanu, Stijn Baert.

**Formal analysis:** Philippe Sterkens, Stijn Baert.

**Funding acquisition:** Stijn Baert.

**Investigation:** Philippe Sterkens, Victor Van Driessche, Michael Geamanu.

**Methodology:** Philippe Sterkens, Victor Van Driessche, Michael Geamanu, Stijn Baert.

**Project administration:** Philippe Sterkens, Ralf Caers, Marijke De Couck, Victor Van Driessche, Michael Geamanu.

**Resources:** Ralf Caers, Marijke De Couck, Stijn Baert.

**Supervision:** Ralf Caers, Marijke De Couck, Stijn Baert.

**Writing – original draft:** Philippe Sterkens, Victor Van Driessche, Michael Geamanu, Stijn Baert.

**Writing – review & editing:** Philippe Sterkens.

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
