## [Decision Letter · Decision Letter 0]

15 Nov 2022

PONE-D-22-19619Costly Mistakes: Why and When Spelling Errors in Resumes Jeopardise Interview ChancesPLOS ONE

Dear Dr. Sterkens,

Thank you for submitting your manuscript to PLOS ONE. After careful consideration, we feel that it has merit but does not fully meet PLOS ONE’s publication criteria as it currently stands. Therefore, we invite you to submit a revised version of the manuscript that addresses the points raised during the review process.

Specifically, the paper deals with an interesting and relevant topic in a rigorous way, but both the experimental setting and external validity could be elaborated more in detail. Furthermore, I encourage to put more effort in providing an explanation for some of your results (such as heterogeneous effects by gender). The three Reviewers provide other comments and suggestions that you should consider in revising your paper.

We look forward to receiving your revised manuscript.

Kind regards,

Federica Maria Origo

Academic Editor

PLOS ONE

Journal Requirements:

2. You indicated that ethical approval was not necessary for your study. We understand that the framework for ethical oversight requirements for studies of this type may differ depending on the setting and we would appreciate some further clarification regarding your research. Could you please provide further details on why your study is exempt from the need for approval and confirmation from your institutional review board or research ethics committee (e.g., in the form of a letter or email correspondence) that ethics review was not necessary for this study? Please include a copy of the correspondence as an ""Other"" file.

No - The funders had no role in study design, data collection and analysis, decision to publish, or preparation of the manuscript.

Reviewers' comments:

Reviewer's Responses to Questions

**Comments to the Author**

1. Is the manuscript technically sound, and do the data support the conclusions?

Reviewer #1: Yes

Reviewer #2: Yes

Reviewer #3: Yes

2. Has the statistical analysis been performed appropriately and rigorously? 

Reviewer #1: Yes

Reviewer #2: Yes

Reviewer #3: Yes

3. Have the authors made all data underlying the findings in their manuscript fully available?

Reviewer #1: No

Reviewer #2: Yes

Reviewer #3: No

4. Is the manuscript presented in an intelligible fashion and written in standard English?

Reviewer #1: Yes

Reviewer #2: Yes

Reviewer #3: Yes

5. Review Comments to the Author

Reviewer #1: This paper assesses the effect of spelling errors on the interview chances of applicants. The authors conducted a vignette online experiment with real recruiters as participants. To obtain causal estimates they randomly allocated spelling errors to job applications. Results show an interview penalty that is stronger for blue-collar jobs and in case of a higher number of mistakes. The paper is well-written and competently executed. Here are my main comments.

1. The authors rely on a vignette experiment to assess the hiring behavior of recruiters. This opens the question of external validity i.e. do we expect the recruiters to behave similarly in a controlled setting observed by the researchers compared to a real job hiring setting? I would appreciate a discussion on this and also what the previous literature said. An alternative more real-world experiment that they could have done is to send fictitious CVs to real job vacancies. Therefore, it is relevant to compare the pro and cons of the vignette study over the CV experiment and whether these two methods tend to produce compatible results.

2. After cleaning the dataset, the authors remain with 445 professionals in their samples. What is the response rate to the online survey and the selection/cleaning rate? Can this affect the external validity of the finding?

3. The authors have other control variables that could be used to enhance the efficiency of the estimates (e.g., participant mother tongue, nationality, education level, and hiring tenure). It seems to me that only for a subset of covariates was used in the final regression (age, gender, language sensitivity), I would therefore like to see the results if all the control variables are included.

4. It should be specified what "α" is (e.g. line 344) as it is not mentioned until that point.

5. I found it interesting that women tend to be hired more than men (see e.g. Table 4). Since also gender was randomized in the application, this may also be interpreted as causal. Do the authors have an explanation for that, for example the type of jobs chosen (this brings back the question of external validity)?

6. The main outcome is "the probability they would invite an applicant for a first job interview (hereafter referred to as ‘interview probability’)" (line 290). I have therefore not clear why the coefficient of B=-1.85 is translated to -18.5 percentage points.

7. The authors compare their results on the total penalty to Toorenburg et al. (2015), which uses white collar and five errors only. The authors should compare Toorenburg to their own estimates on white collar and five errors only.

8. I would clarify the sentence at line 401, by modifying it as follows (or something similar): "As participants could vary in unobserved, confounding variables, a causal interpretation of the interactions regarding the PARTICIPANT CHARACTERISTICS is inappropriate."

9. The authors mention at the beginning of the paper that one of their main contributions is to estimate the effect also for blue-collar occupations since the previous literature mostly focused on white-collar jobs. I would stress throughout the paper that when they mentioned “jobs with high educational requirement" they actually mean white-collar jobs.

Small comments:

- missing space on page 13, line 275

Reviewer #2: This is a well-written research article dealing with an interesting topic. Both the experimental design and empirical analysis are reasonable, and the results are clear.

Some suggestions for improvements:

1. The authors introduce spelling errors in the fictitious resumes. However, language errors come in many forms and could be both pure spelling errors and more structural grammatical errors. It would be nice if the authors could elaborate more on their choice of spelling errors. How common are the spelling errors used? Will standard word-processing software detect these spelling errors, and hence make them less likely to occur in real-world resumes? Would an alternative design with grammatical error be more interesting? The authors should discuss these issues briefly in the main text, and include the actual spelling-errors used in an online appendix. The authors should also include an example of the vignette in the online appendix.

2. The presentation of the results could be improved. Table 4 contains the main results, but this table is rather large and somewhat difficult to read. I would suggest that the authors include more guidance to the reader in the main text about where the reader should look in the table (i.e., clearer references to the panels and columns when they discuss different estimates/results). Also, the results for the interactions with two spelling errors should be included in an online appendix.

3. A major limitation with the study is the hypothetical nature of the methodology used; i.e. the participants know that they answer a survey and that their answers have no real-world consequences. This is mentioned and discussed in the concluding section, but this limitation is so important that it should be highlighted earlier (i.e., briefly in the introduction and in the section on method).

Reviewer #3: I found this paper to be a very compelling and clear read. The authors tackle an interesting question in a very competent and thorough manner. I found some grammar mistakes : ) but they were too few to remember. I would encourage them to pursue a field experiment as a complement to the lab experiment described in their work as it would provide external validation to their results. But that would be the goal of another paper.

I have one small suggestion: to describe how long it took the recruiters to evaluate their three applications and complete the follow-up survey.

6. PLOS authors have the option to publish the peer review history of their article (what does this mean?). If published, this will include your full peer review and any attached files.

Reviewer #1: No

Reviewer #2: No

Reviewer #3: No

---

## [Author Response · Author response to Decision Letter 0]

6 Feb 2023

Dear editor,

Dear reviewers,

Thank you for the opportunity to revise our study for PLOS ONE. We have responded to each of your comments in the files uploaded for the submission.

Thank you for your time and effort.

Kind regards,

The authors

---

## [Decision Letter · Decision Letter 1]

6 Mar 2023

Costly Mistakes: Why and When Spelling Errors in Resumes Jeopardise Interview Chances

PONE-D-22-19619R1

Dear Dr. Sterkens,

We’re pleased to inform you that your manuscript has been judged scientifically suitable for publication and will be formally accepted for publication once it meets all outstanding technical requirements.

Kind regards,

Federica Maria Origo

Academic Editor

PLOS ONE

Additional Editor Comments (optional):

Reviewers' comments:

Reviewer's Responses to Questions

**Comments to the Author**

1. If the authors have adequately addressed your comments raised in a previous round of review and you feel that this manuscript is now acceptable for publication, you may indicate that here to bypass the “Comments to the Author” section, enter your conflict of interest statement in the “Confidential to Editor” section, and submit your "Accept" recommendation.

Reviewer #1: All comments have been addressed

Reviewer #2: All comments have been addressed

2. Is the manuscript technically sound, and do the data support the conclusions?

Reviewer #1: Yes

Reviewer #2: Yes

3. Has the statistical analysis been performed appropriately and rigorously? 

Reviewer #1: Yes

Reviewer #2: Yes

4. Have the authors made all data underlying the findings in their manuscript fully available?

Reviewer #1: Yes

Reviewer #2: Yes

5. Is the manuscript presented in an intelligible fashion and written in standard English?

Reviewer #1: Yes

Reviewer #2: Yes

6. Review Comments to the Author

Reviewer #1: The authors have answered all my comments and questions competently. I look forward to see the publication. Congratulations.

Reviewer #2: (No Response)

7. PLOS authors have the option to publish the peer review history of their article (what does this mean?). If published, this will include your full peer review and any attached files.

Reviewer #1: No

Reviewer #2: No

---

## [Editor Report · Acceptance letter]

15 Mar 2023

PONE-D-22-19619R1 

Costly Mistakes: Why and When Spelling Errors in Resumes Jeopardise Interview Chances 

Dear Dr. Sterkens:

I'm pleased to inform you that your manuscript has been deemed suitable for publication in PLOS ONE. Congratulations! Your manuscript is now with our production department. 

Kind regards, 

on behalf of

Dr. Federica Maria Origo 

Academic Editor

PLOS ONE